# The Impact of Laser Radiation on Polypropylene Molded Pieces Depending on Their Surface Conditions

**DOI:** 10.3390/polym11101660

**Published:** 2019-10-11

**Authors:** Piotr Czyżewski, Dariusz Sykutera, Marek Bieliński, Adam Troszyński

**Affiliations:** UTP University of Science and Technology, Faculty of Mechanical Engineering, Prof. S. Kaliskiego 7, PL 85796 Bydgoszcz, Poland; sykutera@utp.edu.pl (D.S.); biel@utp.edu.pl (M.B.); adatro000@utp.edu.pl (A.T.)

**Keywords:** laser marking, polypropylene, surface finishing, laser marking masterbatch, colorimetry

## Abstract

The article presents an analysis of the impact of a laser beam interaction (Nd: YVO_4_) with selected operational parameters on the quality of graphical marks obtained on the surfaces of polypropylene-molded pieces with different surface textures (variable parameters of the surface layer). Polypropylene test specimens were produced by injection using the original injection mold, which allowed for the obtainment of products with variable surface finish parameters determined by the surface condition of the forming cavity. The presented relationship between the parameters of laser performance, the texture of a molded piece surface, the molded piece’s color, and types of masterbatches supporting the marking process allows for the assessment of the efficiency of graphic symbol application by laser marking. The original evaluation criteria for the conducted process were adopted.

## 1. Introduction

The increasing requirements concerning the quality of the marking of polymer products have been attributed to a need for increasing the durability of the graphic record, as well as to a tendency to individualize products. A use of a permanent marking performed on products is closely related to the development of methods to monitor production lines and is also in agreement with the development of the industry 4.0 concept. Currently, laser marking is used in the production of packaging, cables and technical products in medicine and other everyday-use devices [1,2].

Another important argument for using this method is environmental protection. Product marking by laser radiation reduces the amount of material used in traditional methods of the decorating of product surface layers, such as technologies of printing, painting, tamponing (a specific type of paint), and in IML (in-mold-labelling of foil). This facilitates an effective identification of polymers in the sorting stage [3]. The advantages and disadvantages of laser marking over other methods of surface marking of polymer parts, including an increased susceptibility to recycling, have been described by Kremer and Hahn [4]. A great progress in the development of the laser marking technology has been achieved through the interdisciplinary cooperation of laser manufacturers, the producers of plastics and additives, and selected groups of plastics processors. The Alaska Project, implemented by RWTH Aachen University and the Fraunhofer Institut, is a good example of such a cooperation [5,6].

Laser marking is mainly based on the discoloring of the surface of an element. Short pulses of focused laser radiation cause changes in surface features in a small, local area of a sample. Depending on the type of material, this generates internal stresses, initiating the cracking and deformation of products. A more complex process of changes occurs in composite materials, where it is often necessary to increase laser sensitivity by adding auxiliary materials (additives) [7]. Żenkiewicz et al. extensively described the goals and methods of modifying the surface layer of polymers using laser energy [8]. In turn, Tan and et al. described the use of a laser to form polymer microelements through their precise laser melting. These studies were supported by a numerical analysis of thermal effects in the area of laser operation [9]. Paper [10] presented the results of research on the use of a laser to support the process of polylactide (PLA) surface modification with a copper layer. However, there have only been a few works describing a relationship between a tool surface condition and the quality of the obtained laser record [11].

A color change can be obtained both in the pigment and in the polymer material [12,13]. The interaction of atmospheric oxygen with molten material often produces oxides whose colors provide even better contrast and readability of a graphic symbol. However, the color range in laser marking is significantly limited and depends on the materials that have been used. Already known polymer processing types, such as co-extrusion or two-component injection moulding, allow for additional color effects by removing a layer of one of the polymers [1,7]. Control over the geometrical features of the obtained sign (size, shape, orientation, brightness, etc.) can be acquired by changing the focal field, wavelength, duration, and energy of laser pulses, as well as by changing the temperature of the marked objects [3,14,15]. The mechanism of the absorption of laser radiation depending on its intensity is described by Gnatyuk et al. [3].

A basic parameter for describing the quality of marking is the color contrast between the marked surface and the background. The contrast is not always achieved by the color change itself, as it can be also obtained by the variable light diffraction. This is due to the differences in the geometrical structure of the scanned area and the background (a surface condition).

The color change also occurs as a result of ablation, carbonizing, and a removal of a polymer layer. A methodology for selecting indicators that enable the assessment of the effects of laser marking has been described in numerous research papers. This methodology is mainly based on the use of HPS electron spectroscopy [1,16], Fourier transform infrared spectroscopy (FTIR) [7], scanning electron microscopy (SEM) [7,14] and confocal microscopy [2]. To assess surface conditions, measurements are frequently made with a use of a profilographometer [10,17], while a visual color effect is verified using colorimeters [18,19]. In research using infrared spectroscopy, an increase in the quality of the labelling of specimens with a Sb_2_O_3_ content in a polymer matrix has been confirmed [3]. In the research of [20], to improve the contrast of the laser marking on a polypropylene (PP) product surface, a mixture of a thermoplastic elastomer (TPE) with an addition of titanium dioxide (TiO_2_) was prepared, the latter of which had a significantly improved the laser effect.

In transparent materials, owing to their optical properties, a heterogeneous absorption of radiation occurs (i.e., inclusions and density fluctuations) [5]. Dark-colored polymers, on the other hand, are treated by surface foaming. In this process, the base material is partially melted and gasified. The resulting gas bubbles are stuck on the surface, scattering the incident light. The black inclusions observed on the surface of the material result from the formation of carbon products due to polymer matrix pyrolysis [5,20]. They have been well identified in colorimetric measurements [4,21]. In the case of laser marking additives (LMA), it is recommended to verify their dosage level, because of the costs and the quality of marking. The amount of additional materials from this group to be applied is usually recommended by producers, but it also needs to be verified in real processing conditions [2,18,22]. The impact of the so-called intelligent additives that improve laser marking velocity and the contrast of the obtained graphics was further described in [22,23].

The efficiency of the laser marking of plastics is closely related to the structure of polymers. Some of them are characterized by a relatively high susceptibility to labelling (PS, PC, PVC, PET). However, in the case of polymers such as PE, PP or TPU, it is recommended to use auxiliary materials supporting this process (i.e., laser marking additives—LMA) [14,15]. In the case of PC, these additives can significantly increase contrast, edge sharpness, and marking velocity [17]. In turn, marking of styrene polymers such as ABS, PS, SB and SAN takes place without the use of additives because surface defects, such as dots [12,18], may occur. An increase of the roughness of PET and PTFE surfaces and its hydrophobicity for microbiological applications in extreme ultraviolet (EUV) can also be found [24,25]. The surfaces of PLA products have also been modified by lasers [10].

Both high density polyethylene (HDPE) and polypropylene (PP) belong to the group of materials that poorly absorb radiation with a wavelength of 1064 nm. This is the reason why they hardly undergo any color changes and are not carbonized. Therefore, the efficiency of the marking of the polyolefin plastics is determined by such factors as the degree of surface finishing of a specimen before the marking process and the additives applied to a polymer matrix. The products with a higher roughness generally absorb more laser radiation energy. However, this may well lead to the thermal damage of a surface layer. A smooth surface allows for homogeneous processing, but it is characterized by a higher radiation reflection coefficient [3,13,15]. From the above, it seems reasonable to examine the impact of the surface condition of polymer-molded pieces formed in injection mold cavities on the effectiveness of laser marking, particularly for materials such as PP. The research of [2] confirmed a positive effect of a selected LMA additive on the quality of the marking of PP white samples. Simultaneously, it was found that in the case of black samples made of the same type of PP and modified with the same additive, the marking effect is quite the opposite. This indicates a different interaction between the color concentrate and the LMA.

The aim of the study was to assess the effectiveness of laser radiation impact on polypropylene-molded surfaces with various geometrical features. It was assumed that depending on the method of finishing of the injection molding cavities, the variable parameters of laser marking, and the amount of the LMA-absorbent additive, different marking effects could be expected to be obtained on molded pieces with different levels of roughness. The research included the production of PP specimens by injection molding, the identification of the geometrical features of these specimens’ surfaces, the conduction of laser marking tests, and the determination of the changes in color and geometric condition of molded surfaces in the area of marking (in regards to the above-mentioned background information).

## 2. Methodology

For the research, Moplen HP 500N polypropylene (Bassell Orlen Polyolefins, Płock, Poland) was used. The Schwarz 501 PE coloring concentrate content of 2 wt% (Lifocolor Farben GmbH & Co. KG, Lichtenfels, Germany) was added to the original granulate. It contained a carbon black content of 19 wt% and a calcium carbonate content of 51 wt%. In addition, the PP granulate was modified with an LMA additive, i.e., Lifolas M Schwarz 113504 UN (Lifocolor Farben GmbH & Co. KG, Lichtenfels, Germany). It was added to polypropylene in contents of 0 wt%, 0.5 wt%, 1.5 wt% and 2.5 wt%.

Due to the purpose of the research, the test specimens were produced by high-pressure injection in a special, modular injection mold (Figure 1a). Its original design allowed for the obtainment of flat samples with dimensions of 108 mm × 94 mm × 2 mm. The modularity of this solution included the ability to quickly replace the steel forming insert. The forming part was divided into three fields with the same surface but differing in the degree of a surface finishing. In the case of the first insert, forming surfaces were obtained by polishing (marked with P), grinding (the central part of the insert—marked with S), and honing (marked with O). The second insert was manufactured by EDM (electrical discharge machining), with three separate settings adopted for this process. As a result, the insert was characterized by three different forming surfaces of various levels of roughness (marked with DA; DB, which was the middle part; and DC). Due to such a design elements, flat specimens were produced, and three fields with variable topography could be identified on the surface of each specimen (Figure 1b). This resulted from the nature of the injection molding, in which the molten material was transported into the mold with high pressure, and it reproduced the shape and the surface of an injection mold cavity as negatives.

The Battenfeld Plus 350/75 injection molding machine (Battenfeld Kunststoffmaschinen GmbH, Kottingbrunn, Austria) was used to produce test specimens. Table 1 presents parameters set to prepare the specimens. A good homogenization of additives was obtained by pre-mixing the ingredients in a plasticizing system of the W25-30D extruder with a screw diameter of 25 mm (Metalchem, Toruń, Poland). Table 2 presents the parameters used to prepare the granulate. The resulting extrudate was granulated in a cold temperature. Each time, after modifying the processed mixture with a different content of Lifolas M Schwarz 113504 UN, the plasticizing unit of the injection molding machine and the extrusion machine was cleaned using the PP Moplen HP 500 N basic plastic.

The surface conditions of the forming insert and PP-molded pieces were determined by studying the surface topography on a MarSurf GD 120 measuring device (Mahr GmbH, Göttingen, Germany). To prepare the characteristics of the geometrical features of cavities and test specimens, the following three parameters were selected: Two-dimensional R_a_, three-dimensional arithmetic surface deviation S_a_, and scale VDI 3400 reference (commonly used to describe the surface conditions of thermoplastic products). Figure 2 presents exemplified images of the surface topography of the research samples for each type of surface. Table 3 presents the results, which constituted a starting point for the implementation of the experiment and allowed for a precise analysis of the obtained results.

Graphic signs with the dimensions of 20 mm × 18 mm were applied on the surfaces of test samples using the TruMark Station 1000 laser marking machine (Trumpf Group, Ditzingen, Germany) equipped with a 1064 nm Nd:YVO_4_ laser. The device was controlled by the TruTropsMark software (Trumpf Group, Ditzingen, Germany). Table 4 illustrates laser marking parameters used to prepare the research samples. Figure 3 presents images of graphic signs obtained by the laser beam impacting on the surfaces of molded pieces with various levels of roughness, together with the adopted values of the laser marking velocity.

The colors of the specimens were measured using a Ci6X colorimeter (XRite, Grand Rapids, MI, USA). The CIELAB system was used to determine the differences in colors of the surfaces before and after the marking. As the criterion to assess the efficiency of the laser marking performed on the PP samples, the following characteristic parameters of this color measurement system were selected: Δ*L**, brightness difference, and Δ*E**, the general color deviation (according to DIN 6174). This choice was made due to the fact that it is the difference in color and the contrast between the laser graphic sign and the remaining surface of the molded piece that determine the effectiveness of laser marking. Five replications of the color measurement were carried out on each of the marked surfaces.

## 3. Results

For the analyzed ranges of the laser head velocity, the largest difference in color and luminance resulting from the laser beam impact on the surface of the molded pieces was obtained for the samples of Moplen HP 500N polypropylene without any content of Lifolas M Schwarz 113504 UN. The changes in Δ*E** for the PP specimens with variable surface finishings ranged from approximately 18.5 to 34.5 (Figure 4a). In the case of these samples, the desired effect of a color change on the surface of a laser-marked molded piece was obtained.

The increase in the Δ*E** parameter value in the CIELAB space effects primarily from a visible improvement of the brightness L, but also from the color changes towards the red in the “*a*” axis and towards the yellow color in the “*b*” axis (see Table 2). Regardless of the type of a product surface, the major color change is characteristic for the samples marked with the laser head velocity B (750 mm·s^−1^) (see Figure 4a). Simultaneously, the greatest impact of the PP product surface condition without any marker additive on the changes in Δ*E** and Δ*L** was observed for marking velocity A (450 mm·s^−1^). For the samples characterized by surfaces with low R_a_ and S_a_ values (i.e., marked by P, S and O), it was found that the lower the gloss of the forming cavity and the sample itself, the more visible the contrast effect between the laser-marked surface and its surrounding area. For the molded pieces obtained in EDM cavities, the R_a_ increase of the molded surfaces resulted in an increase of Δ*E** for the marked surfaces with the lowest velocities of the head movement.

The increase in the laser beam interaction velocity to C (1050 mm·s^−1^) and D (1350 mm·s^−1^) values resulted in marked surfaces that differ in a shade of color from the remaining area to a lesser extent than for the B variant. The exceptions were the DA surface, for which the Δ*E** parameter exceeded a value of 32 (see Figure 4a). These unfavorable changes resulted from a decrease in the luminance value, which remained at a higher level than the pattern’s (the Δ*L** value was above 20) and from the smaller values of Δ*a** and Δ*b**, which still maintained positive values (value changes in the direction of the a-axis are shown with the red color, and value changes of the b-axis are shown with the yellow color), as can be seen in Table 5.

To sum up this part of the research, it can be concluded that all changes in the color of the marked surface for the PP samples without the addition of a marker resulted from a significant increase in Δ*L** ranging from 17.57 to 31.01 for the DB surface and, to a lesser extent, from positive changes in the values of the a-axis components (from 0.64 for the S surface to 4.26 for the DB surface), as well as in the range from the value of +4.48 (P surface) to the value of +16.99 (DB surface) in the b-axis (see Table 5). Changes in the color saturation (chroma), depending on Δ*a** and Δ*b**, were minor when compared to the referential surfaces for the highest velocity of laser head movement. They were major when the laser head operated at the lowest velocity (see Table 5). As a result, for the PP samples without LMA, the images of marked fields were significantly brightened in warm tones with shades of colors from beige to brown (see Figure 5).

The addition of the Lifolas M Schwarz 113504 UN marker worsened the marking effects, regardless of the surface type (see Figure 4b). This was evidenced by the values of Δ*E**, which decreased when compared to the reference surfaces of the molded pieces before the marking, mainly due to smaller differences in Δ*L** but also due to the decrease in the values of the Δ*a** and Δ*b** parameters (Table 5). When these changes were even more visible and less advantageous, the content of the marker was greater (see Figure 6). However, it should be added that the brightening effect of the marked surface in relation to the surrounding area was still maintained. The exceptions were the S-type molded pieces with the largest LMA contents, for which the interaction of the laser with the velocity C caused a negative increase in Δ*L**. The lowest values of the Δ*E** parameter occurred in the case of the PP specimens with the 2.5 wt% Lifolas M Schwarz 113504 UN content and marked with the lowest laser head velocity (see Figure 4b and Figure 6). The impact of the laser beam on a molded part surface over a longer period of time (at a lower velocity) causes its degradation in a form of carbon compounds. This effect was further enhanced by the addition of 2 wt% PP black dye with significant amounts of carbon black (content of about 19 wt%) and CaCO_3_ content of as much as 51 wt%, as can be seen in Figure 7.

In some of the cases of the laser beam impact on molded pieces with velocity A, defects in a form of deep notches were observed. Thus, the recorded images were called the plowed soil (see Figure 8). The tendency for unfavorable changes in Δ*E** could be observed for all types of surfaces of molded pieces. As a part of these changes, the use of the velocity B for the samples with the 0.5 wt% Lifolas M Schwarz 113504 UN content and velocity D in the case of PP-molded pieces with 1.5 wt% and 2.5 wt% contents allowed for the highest values of Δ*E** and Δ*L**. Additionally, in these cases, it can be observed that the most favorable values of these parameters were obtained for the molded pieces produced by EDM, particularly for the DC with the highest roughness value. It was also acknowledged that once the laser beam had passed, marked surfaces with high roughness (i.e., DA, DB, DC) were smoothed (Figure 7).

An analysis of the surface condition of PP samples after the laser marking process clearly indicated that the surface condition before the marking of the molded piece affected the marking results to a considerable extent. In the case of the molded part with a high degree of surface finishing (i.e., with low roughness), both the R_a_ and S_a_ parameters significantly deteriorated (Table 6). These could be compared to the surface of molded parts before marking, obtained through the forming of cavities made by EDM. High CH index (Charmilles) values according to the VDI 3400 scale indicated significant damage on the surfaces of molded pieces. This tendency was intensified by the addition of LMA to PP, with a major change observed for the samples with the 2.5 wt% Lifolas M Schwarz 113504 UN additive content. In the case of high-roughness molded pieces (i.e., the DA, DB, and DC surfaces), it was observed that the R_a_ and S_a_ changes in the surface condition of the samples before the marking were much smaller; for the DB and DC samples not containing LMA, the impact of laser radiation improved these surface parameters (see Table 6). Similarly to the surface specimens of P, S and O, the addition of a marker deteriorated the surface topography in the laser field. Examples of the effects of laser radiation on PP samples are shown in Figure 9. On the topographic maps presenting injection molded samples, it can be observed that the laser marking process eliminated or significantly limited the surface changes, reflecting traces of the surface treatment of the injection molding insert in the injection mold (see Figure 2 and Figure 9).

## 4. Conclusions

The research confirmed that polypropylene is a material whose susceptibility to laser marking is conditioned by a much larger number of factors than in the case of other construction plastics. It was found that the best labelling effects were obtained for polypropylene specimens with the addition of Schwarz 501 PE coloring at a concentration of 2 wt%. In this case, an additional modification of the material with the LMA Lifolas M Schwarz 113504 UN worsened the obtained effects of color and contrast change in the areas of laser interaction on the surface of a molded piece. Considering the results of the presented research and our own previous study [2], it can be stated that achieving the right contrast between the laser graphic symbol and the remaining surface of the black-colored polypropylene product depends on the composition of the color concentrate and other additives (e.g., talc, chalk, and LMA). Therefore, it was confirmed that laser marking of polyolefin products is more difficult and occurs as a result of the mutual interaction of both the carbonization process and porosity in the surface layer of the product. The latter mechanism particularly allows the material (e.g., CaCO_3_ or talc) with a higher temperature to be exposed. Hence, the preparation of an appropriate mineral supplement can give other interesting visual effects.

It was found that both the surface of the sample being marked and the velocity of the laser head have a significant impact on the color change of the graphic symbol obtained through the interaction of the laser beam on the PP surface. The best effects were obtained using the B velocity for the specimens without the addition of LMA and for the velocity of C and D for the molded parts containing the Lifolas M Schwarz 113504 UN.

Subsequent research will determine the impact of the mold temperature on surface condition and its relationship to the parameters of the laser marking process and the LMA content in the polypropylene matrix.

## Figures and Tables

**Figure 1 polymers-11-01660-f001:**
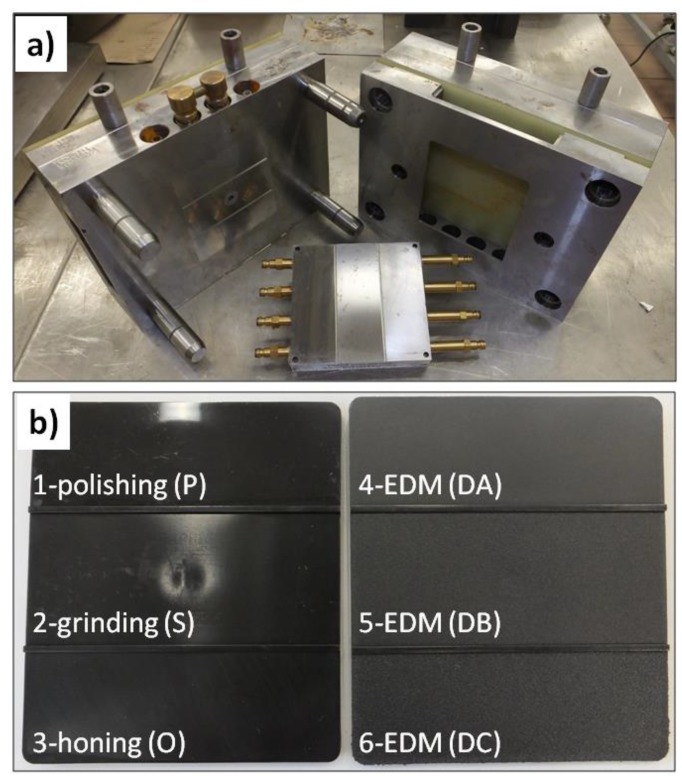
View of a modular injection mold with a visible forming insert and a cooling system (**a**), allowing for the production of the molded specimens with three characteristic areas of different surface topography parameters (**b**).

**Figure 2 polymers-11-01660-f002:**
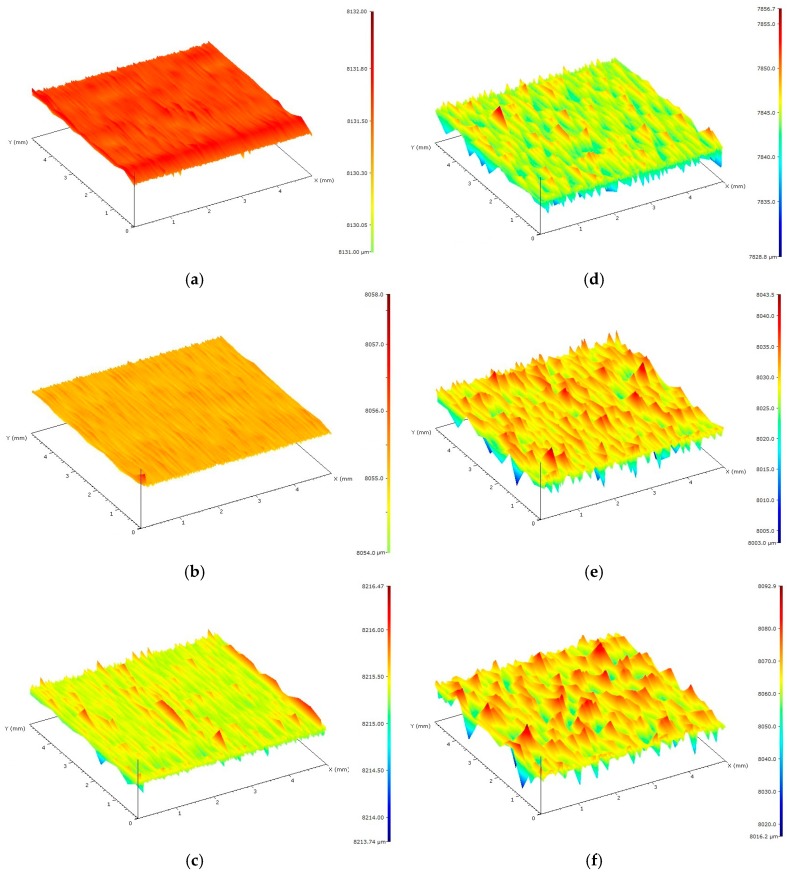
Sample images of surfaces of polypropylene (PP) specimens used in the research: (**a**) The polished one (P), (**b**) the ground one (S), (**c**) the honed one (O), (**d**) the electrical discharge machining (EDM)(DA) one, (**e**) the EDM (DB) one, and (**f**) the EDM (DC) one.

**Figure 3 polymers-11-01660-f003:**
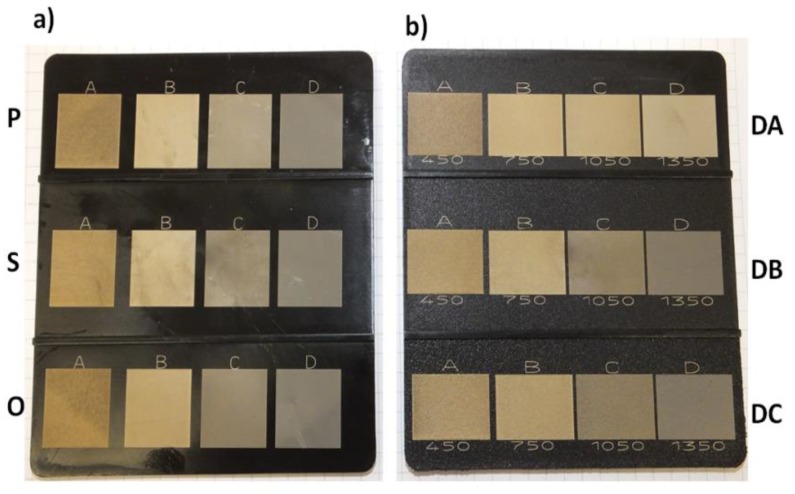
A graphic representation of the ways to modify the surfaces of the plates with the references to the adopted laser parameter (a plate on the right). (**a**) The surfaces: The polished one (P), the ground one (S), and the honed one (O). (**b**) The surfaces: The EDM (DA) one, the EDM (DB) one, and the EDM (DC) one.

**Figure 4 polymers-11-01660-f004:**
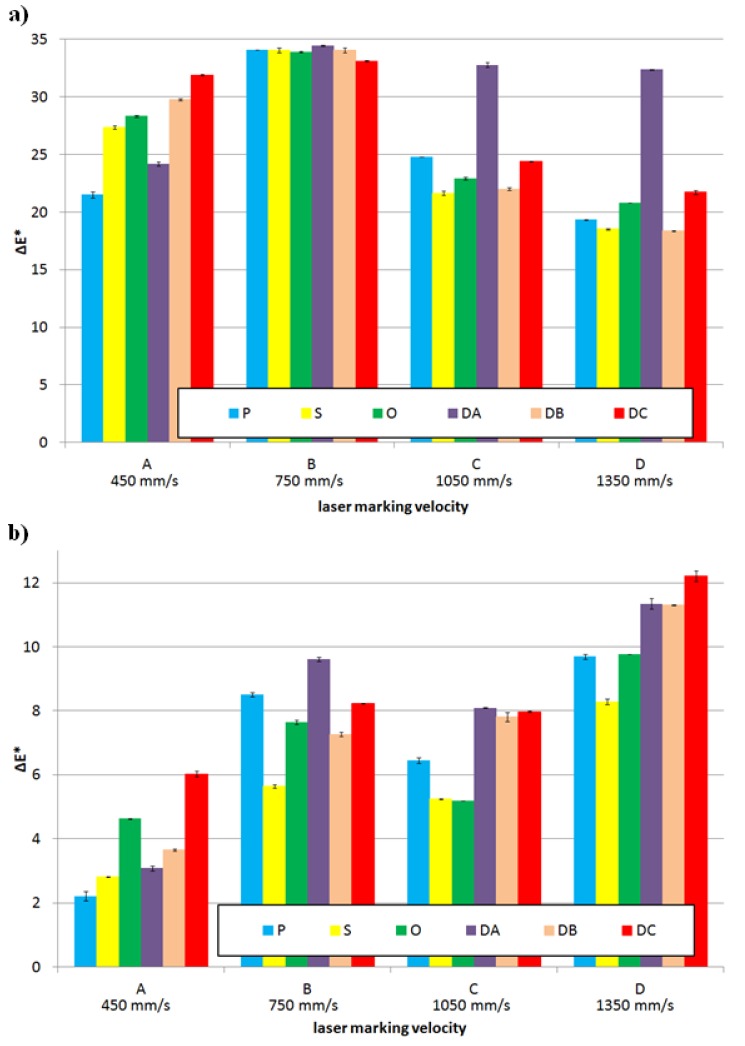
Influence of the additive content absorbing laser energy on the Δ*E** (general color deviation) parameter in relationship to a surface used and a laser marking velocity: (**a**) 0.0 wt% of the LMA additive content and (**b**) 2.5 wt% of the LMA additive content.

**Figure 5 polymers-11-01660-f005:**
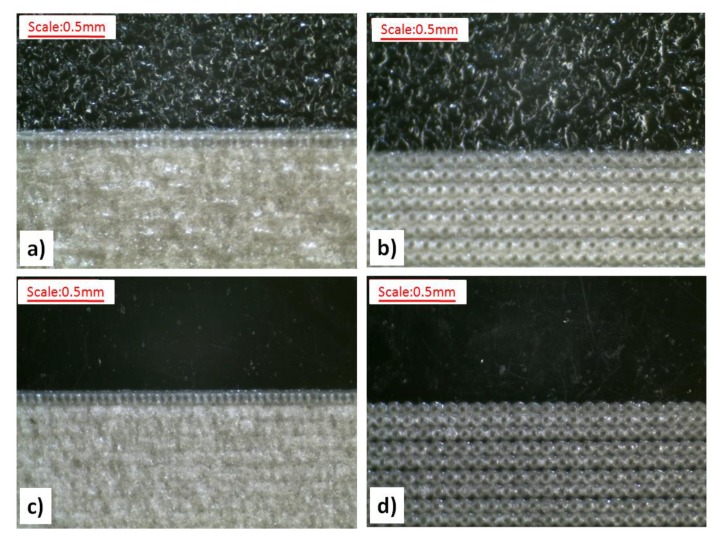
The influence of laser marking velocity on the exemplary surfaces of samples without LMA: (**a**) DA surface, B laser head velocity (750 mm·s^−1^); (**b**) DA surface, D laser head velocity (1350 mm·s^−1^); (**c**) P surface, B laser head velocity; and (**d**) P surface, D laser head velocity.

**Figure 6 polymers-11-01660-f006:**
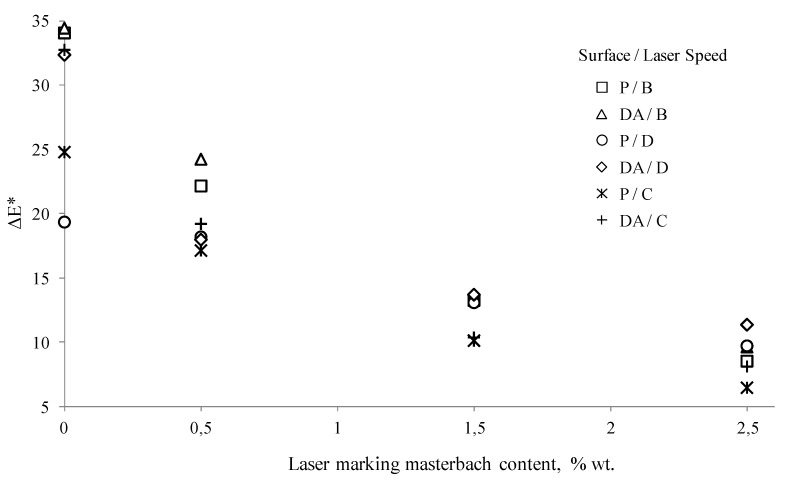
Influence of the laser marking masterbatch content on the value of the Δ*E** for the P and DA surfaces, obtained with the laser marking velocity of B (750 mm·s^−1^), C (1050 mm·s^−1^), and D (1350 mm·s^−1^).

**Figure 7 polymers-11-01660-f007:**
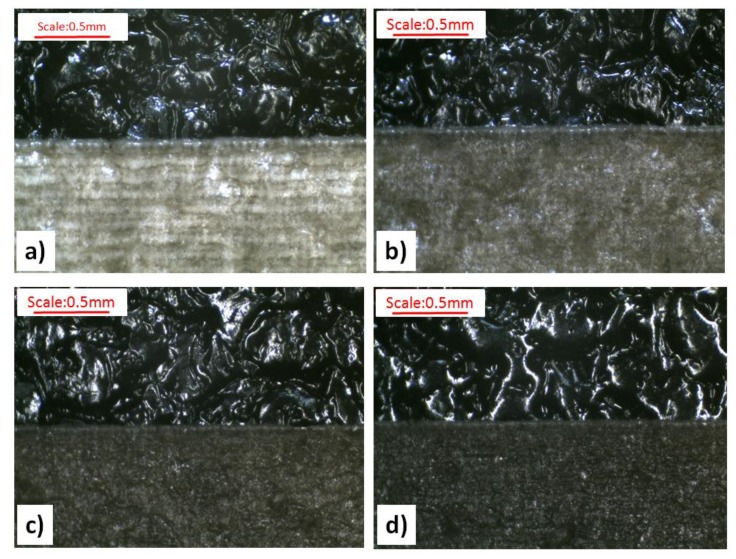
Influence of the Lifolas M Schwarz 113504 UN laser marking additive content on the condition of the DC surface before and after the marking with velocity B (750 mm·s^−1^): (**a**) 0.0 wt% content, (**b**) 0.5 wt% content, (**c**) 1.5 wt% content, and (**d**) 2.5 wt% content.

**Figure 8 polymers-11-01660-f008:**
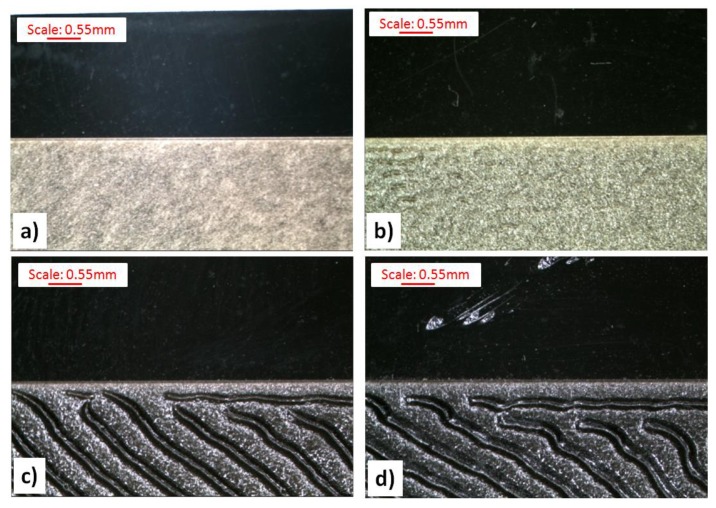
Influence of the Lifolas M Schwarz 113504 UN additive content on the surface condition of the P surface before and after the marking at velocity A (450 mm·s^−1^): (**a**) 0.0 wt% content, (**b**) 0.5 wt% content, (**c**) 1.5 wt% content, and (**d**) 2.5 wt% content.

**Figure 9 polymers-11-01660-f009:**
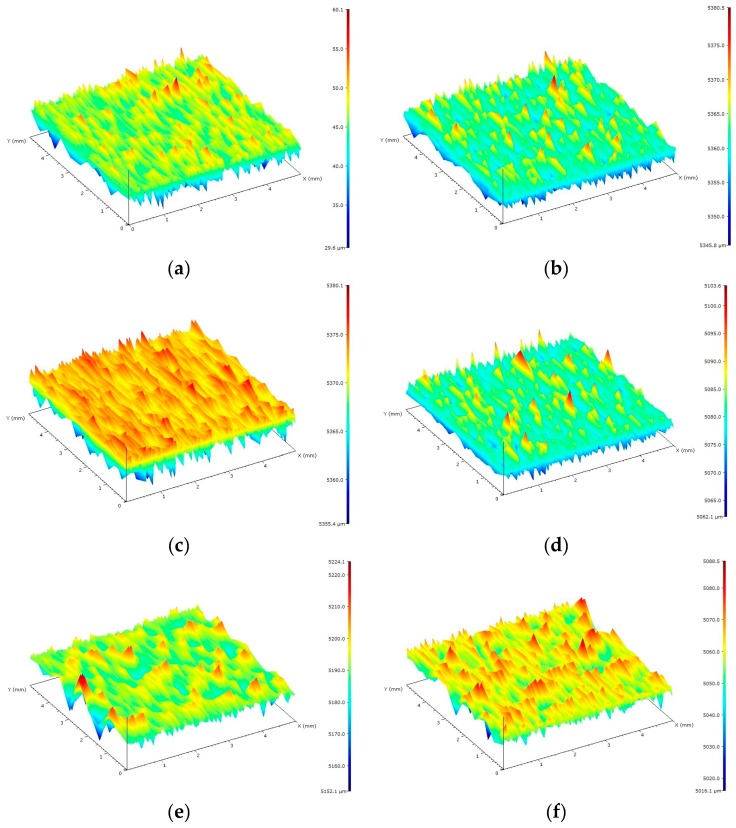
Examples of topography images of the surface area of PP samples after laser marking at velocity B, depending on LMA content and surface type before the marking: (**a**) P surface without LMA; (**b**) P surface with an LMA content of 2.5 wt%; (**c**) O surface without LMA; (**d**) O surface with an LMA content of 2.5 wt%; (**e**) DC surface without LMA; and (**f**) DC surface with an LMA content of 2.5 wt%.

**Table 1 polymers-11-01660-t001:** Injection molding parameters used to prepare samples.

Process Parameter	Value
Injection pressure [MPa]	78.75
Injection time [s]	0.68
Holding pressure [MPa]	8.75/6.56
Holding time [s]	6/6
Temperature of the plasticizing unit zones [°C]	I-200, II-210, III-230
Moldould temperature [°C]	20
Cooling time [s]	16

**Table 2 polymers-11-01660-t002:** Extrusion parameters used to prepare granulate.

Process Parameter	Value
Temperature of the plasticizing unit zones [°C]	I-135, II-180, III-200, IV-200
Extrusion head temperature [°C]	200
Screw rotation speed [rpm·min^−1^]	140

**Table 3 polymers-11-01660-t003:** Surface topography parameters (average values) for the adopted types of the surface finishing of injection molding cavities used in the research and for the molded parts obtained in these cavities. Samples of PP without the addition of laser marking additives (LMA) and a mold temperature of 20 °C.

Insert No.	Surface Type	Marking	Forming Insert			Surface of a Molded Piece		
			Ra [µm]	Sa [µm]	VDI 3400	Ra [µm]	Sa [µm]	VDI 3400
1	Polishing	P *	0.011	0.046	CH 0	0.018	0.050	CH 0
	Grinding	S *	0.296	0.275	CH 9	0.243	0.249	CH 8
	Honing	O *	0.117	0.107	CH 1	0.110	0.093	CH 1
2	EDM	DA *	3.080	2.573	CH 29	3.045	2.137	CH 29
	EDM	DB *	5.860	4.207	CH 35	6.019	3.813	CH 35
	EDM	DC *	16.154	9.476	CH 44	14.088	7.941	CH 43

* In the following part of the paper, the designations of surface types of the forming cavities included in Table 3 determine the types of specimens made of PP obtained with the use of these forming inserts.

**Table 4 polymers-11-01660-t004:** Laser marking parameters used to prepare the samples.

Parameter	Value			
Laser power [W]	5			
Path width [mm]	0.07			
Diameter of single impulse [mm]	0.07			
Pulse frequency [Hz]	1500			
Head velocity [mm·s^−1^]	450	750	1050	1350
Distance between single impulses [mm]	0.03	0.05	0.07	0.09
Description of marked area	A	B	C	D

**Table 5 polymers-11-01660-t005:** Color changes in the marked areas in relation to the remaining surface in axes a and b, depending on the content of the Lifolas M Schwarz 113504 UN filler, a type of surface of the forming insert and the velocity of a laser head.

Specimen Type	Surface Type	Δ*a** *(+R*)				Δ*b** *(+Y*)			
		Head Velocity				Head Velocity			
		A	B	C	D	A	B	C	D
PP 0 wt%	P	2.92	2.59	1.55	0.64	11.64	13.95	8.80	4.48
	S	4.17	2.92	1.46	0.69	16.16	14.89	8.56	5.09
	O	3.80	2.97	1.47	0.70	15.22	14.83	8.35	5.07
	DA	4.15	3.37	2.53	1.46	14.36	16.39	13.4	10.83
	DB	4.26	3.12	1.56	0.73	16.99	15.57	9.04	5.29
	DC	3.77	3.17	1.60	1.46	16.07	15.28	8.98	5.66
PP 2.5 wt%	P	0.64	1.33	1.23	0.76	1.56	4.97	4.54	3.53
	S	0.91	1.53	1.39	0.85	2.46	5.27	5.04	4.19
	O	0.94	1.47	1.21	0.79	2.93	5.22	4.49	3.97
	DA	0.67	1.49	1.27	0.78	2.40	5.92	5.34	4.36
	DB	1.09	1.77	1.38	0.81	3.09	6.08	5.59	4.40
	DC	1.13	1.64	1.24	0.79	3.44	5.66	5.01	4.34

**Table 6 polymers-11-01660-t006:** Topography of PP samples after the laser beam interaction.

Specimen Type	Surface Type	Laser Marking Additive (LMA) Content wt%					
		0%			2.5%		
		R_a_ [µm]	S_a_ [µm]	VDI 3400	R_a_ [µm]	S_a_ [µm]	VDI 3400
Polishing	P	3.311	2.953	CH 30	4701	3.525	CH 33
Grinding	S	3.660	2.972	CH 31	5939	5.475	CH 35
Honing	O	2.900	2.577	CH 29	5003	3.824	CH 34
EDM	DA	3.786	3.132	CH 32	4727	3.761	CH 33
EDM	DB	4.686	3.526	CH 33	8494	5.776	CH 39
EDM	DC	9.509	6.339	CH 40	11,570	7.484	CH 41

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
