# Peer review of "The Impact of Laser Radiation on Polypropylene Molded Pieces Depending on Their Surface Conditions"

_polymers, 2019, doi:10.3390/polym11101660_

Round 1

Reviewer 1 Report

  The authors investigate the Nd:YVO4 laser beam on the quality of laser marking on the polyolefin products. They prepared polypropylene samples with different degree of surface finishing through molding and electrical discharge machining (EDM) process. The effect of laser parameters on the laser marking of polypropylene was analyzed. Moreover, the PP matrix with different laser marking additives was also studied. They conclude that the reasons for laser marking polyolefin products are both the carbonization and porosity in the surface layer. The experiments were carefully carried out, and the research results have a guiding role in the laser marking industry.

  However, this manuscript also has many problems, the following issues and questions need to be addressed:

1) In Figure 2, why did the surface morphology of the injection-molded sample change so much after electrical discharge machining (EDM) process? Please explain.

2) In the laser irradiation process, what were the values of the laser power and the line spacing of laser scanning? How to effectively control the focus of the laser beam? Please provide specific parameters.

3) Scanning electron microscopy (SEM) is a powerful surface topography characterization tool. If possible, I suggest the authors better to conduct the SEM images before and after laser marking.

4) On page 6, Line 169, “…width of a single laser line of 0.07mm…” How did the authors measure the width of a single laser line? What was the spot size of the laser used in the experiments?

5) On page 9, Line 234, “The addition of the Lifolas M Schwarz 113504 UN marker worsened the labeling effects”, why? Please give a simple explanation.

6) In Figure 4, to make it easier to understand, the authors need to point the laser marking speed into the X-axis directly, instead of using ABCD.

7) I found that the authors mark light-colored patterns on black substrates, so the mechanism of laser marking should be foaming of the polymer substrate rather than carbonization. Whether the carbonization really occurred, please give the experimental evidence or corresponding references.

8) The format and writing of this paper need to be standardized.

  For example, on Page 1, Line 19, “polypropylene” instead of “polipropylene”; on Page 2, Line 60, “…on a polypropylene (PP) product…” instead of “…on a PP polypropylene product…”; on Page 2, Line 82, “…surfaces of polylactide (PLA) products have…” instead of “…surfaces of polylactide products (PLA) have…”; on Page 2, Line 84, “…the high-density polyethylene (HDPE) and the polypropylene (PP) belong to…” instead of “…the HDPE high density polyethylene and the PP polypropylene belong to…”; on Page 6, Line 180, “…to assess the efficiency…” instead of “…to asess the efficiency…”; and so on.

9) The paper needs an extensive revision for English grammar.

  In conclusion, this manuscript needs a major revision before the publication in Polymers.

Author Response

Dear Reviewer,

Thank you very much for your comments and inspiring suggestions concerning my manuscript. The corrections and modifications to the manuscript have already been made in accordance with your reccomendations. Below, please kindly find the detailed explanations and answers to your questions:

1) In Figure 2, why did the surface morphology of the injection-molded sample change so much after electrical discharge machining (EDM) process? Please explain.

Ad.1 The methodology for carrying out the experiment was in agreement with the subject in the title of the manuscript. At the same time, the work was to be of a utilitarian nature. That is why the injection moulding surfaces were made with the use the most popular techniques. EDM (Electrical Discharge Machining) is a technique that uses the phenomenon of burning the surface of a material under the influence of an electric arc. An electric arc is created between the working surface and the electrode. For EDM machining of injection mould cavities, process parameters (voltage and current) were selected in such a way as to obtain surfaces with the expected and most popular morphologies. Some of the cavities are made by abrasive machining to obtain moulded pieces with a high degree of surface finishing (high gloss). This type of effect is achieved by finishing (e.g. polishing, honing and grinding). For this reason, the forming cavity was characteristic of 6 different surface conditions.

2) In the laser irradiation process, what were the values of the laser power and the line spacing of laser scanning? How to effectively control the focus of the laser beam? Please provide specific parameters.

Ad.2. The laser parameters used for the tests are summarized in Table 4. The values of the spacing between the lines of the marked area are the result of the values of the linear velocity of the laser head adopted during marking.

3) Scanning electron microscopy (SEM) is a powerful surface topography characterization tool. If possible, I suggest the authors better to conduct the SEM images before and after laser marking.

Ad.3. We agree with the reviewer that SEM images would expand the area of analysis. However, the authors' intention was to attempt to determine the influence of the condition of the surface of forming cavities on the quality of marking of moulded surfaces. Therefore, we decided that the results analyzing the changes on a larger surface of the sample (high quality optical microscope and surface topography analyzer) should be presented firstly. On the other hand, the SEM study covers small areas, which could lead to erroneous conclusions. Nevertheless, this is an interesting suggestion that we will use in further supplementary tests of the obtained samples. However, this requires the development of an appropriate methodology (selection of the measurement field, number of samples, sample preparation, etc.).

4) On page 6, Line 169, “…width of a single laser line of 0.07mm…” How did the authors measure the width of a single laser line? What was the spot size of the laser used in the experiments?

Ad.4. The given value of a single laser line was set in the laser control system. According to the instructions, the expected and repeatable value of the width of a single laser line is obtained by the correct setting of the focal length. After the laser marking, it is not possible to verify the width of the path and the distance between individual characters on the surface, because they overlap (the detailed data are presented in Table No.4).

5) On page 9, Line 234, “The addition of the Lifolas M Schwarz 113504 UN marker worsened the labeling effects”, why? Please give a simple explanation.

In the test, Schwarz 501 PE dye and Lifolas M Schwarz 113504 UN marking additive were added to PP Moplen 500N material. The dye contained a significant amount of calcium carbonate (content 51 wt %), and thus, the best contrast was obtained (removal of the polymer surface layer and the exposition of CaCO3 particles on the labelled surface). The addition of Lifolas M Schwarz 113504 UN disturbed this effect (deterioration of contrast and reduction of ΔE value), which can be explained by the carbonizing effect of this additive.

6) In Figure 4, to make it easier to understand, the authors need to point the laser marking speed into the X-axis directly, instead of using ABCD.

Ad. 6 Marking speed velocities have been supplemented in Figure 4.

7) I found that the authors mark light-colored patterns on black substrates, so the mechanism of laser marking should be foaming of the polymer substrate rather than carbonization. Whether the carbonization really occurred, please give the experimental evidence or corresponding references.

Ad. 7. Based on the current state of knowledge, it is known that the marking of polyolefin materials is a complex process. Both foaming and carbonization take place during the implementation of this process. The intensity of each of them is determined by many processing factors as well as the type of polyolefin modifiers and fillers used in the matrix. In the conducted experiment, in the range of accepted variable factors, the obvious deterioration of the ΔE parameter for samples with the highest content of Lifolas M Schwarz 113504 UN was observed. Since the foaming increases the contrast between the black surface of the product and the sign applied by laser, we have recognized that the deterioration of this effect after the addition is a result of carbonization. In our view, this process results from a better absorption of laser radiation energy on the sample surface. However, we have not provided proof in this respect. This type of analysis will be performed in the subsequent research.

8) The format and writing of this paper need to be standardized.

Ad.8 The descriptions throughout the manuscript have been corrected and standardized.

9) The paper needs an extensive revision for English grammar.

Ad.9 The manuscript was reviewed and the grammar was corrected.

I would be honoured if you consider the manuscript worth publishing in the Polymers The impact of laser radiation on polypropylene moulded pieces depending on their surface conditions now.

Please accept my kind regards,

Piotr Czyżewski

Department of Material Engineering and Polymer Processing

Manufacturing Technologies Institute

Faculty of Mechanical Engineering

UTP University of Science and Technology

Al. Prof. Kaliskiego 7, 85-796 Bydgoszcz, Poland

tel.: +48 52 340 82 22

p.czyzewski@utp.edu.pl

Reviewer 2 Report

The research is well designed and well described but the authors should better highlight the industrial impact of the results obtained.

Moreover here you can find some minor revision (pag:line):

(1:43) - too much citations, please rewrite the sentence and better specify who do what

(2:57) - same revision

(4:133) - to better understand the research add a table to resume per injection process parameter

(6:168) - add the laser power and add a table were all the process parameters investigated are reported

Author Response

Dear Reviewer,

Thank you very much for your comments and inspiring suggestions concerning my manuscript. The corrections and modifications to the manuscript have already been made in accordance with your reccomendations. Below, please kindly find the detailed explanations and answers to your questions:

The research is well designed and well described but the authors should better highlight the industrial impact of the results obtained.

Ad.1. In Conclusions part, the manuscript was supplemented with information on the impact of the obtained results on industry.

Moreover here you can find some minor revision (pag:line):

(1:43) - too much citations, please rewrite the sentence and better specify who do what

(2:57) - same revision

Ad.2. The description of the research was corrected and made more detailed.

(4:133) - to better understand the research add a table to resume per injection process parameter

Ad.3. A table with injection and extrusion process parameters has been added.

(6:168) - add the laser power and add a table were all the process parameters investigated are reported

Ad.4. A table with detailed laser marking parameters has been added.

I would be honoured if you consider the manuscript worth publishing in the Polymers The impact of laser radiation on polypropylene moulded pieces depending on their surface conditions now.

Please accept my kind regards,

Piotr Czyżewski

Department of Material Engineering and Polymer Processing

Manufacturing Technologies Institute

Faculty of Mechanical Engineering

UTP University of Science and Technology

Al. Prof. Kaliskiego 7, 85-796 Bydgoszcz, Poland

tel.: +48 52 340 82 22

p.czyzewski@utp.edu.pl

Round 2

Reviewer 1 Report

In the revised manuscript, my concerns were fully addressed. I would like to recommend the publication of this manuscript.